# Identification of a Pathogenic Mutation for Glycogen Storage Disease Type II (Pompe Disease) in Japanese Quails (*Coturnix japonica*)

**DOI:** 10.3390/genes16080975

**Published:** 2025-08-19

**Authors:** Abdullah Al Faruq, Takane Matsui, Shinichiro Maki, Nanami Arakawa, Kenichi Watanabe, Yoshiyasu Kobayashi, Tofazzal Md Rakib, Md Shafiqul Islam, Akira Yabuki, Osamu Yamato

**Affiliations:** 1Laboratory of Clinical Pathology, Joint Faculty of Veterinary Medicine, Kagoshima University, Kagoshima 890-0065, Japan; faruqabdullahal103@gmail.com (A.A.F.); k6993382@kadai.jp (S.M.); k9829543@kadai.jp (N.A.); rakibtofazzal367@gmail.com (T.M.R.); si.mamun@ymail.com (M.S.I.); yabu@vet.kagoshima-u.ac.jp (A.Y.); 2Faculty of Veterinary Medicine, Chattogram Veterinary and Animal Sciences University, Chattogram 4225, Bangladesh; 3Laboratory of Veterinary Pathology, Department of Veterinary Medicine, Obihiro University of Agriculture and Veterinary Medicine, Obihiro 080-8555, Japanknabe@obihiro.ac.jp (K.W.); kyoshi@obihiro.ac.jp (Y.K.)

**Keywords:** glycogen storage disease type II, Pompe disease, Japanese quail, deletion mutation, *GAA I* gene

## Abstract

**Background/Objectives**: Pompe disease (PD) is a rare autosomal recessive disorder caused by a deficiency of the lysosomal acid α-1,4-glucosidase (GAA) encoded by the *GAA* gene, leading to muscular dysfunctions due to pathological accumulation of glycogen in skeletal and cardiac muscles. PD has been reported in several animals and Japanese quails (JQ; *Coturnix japonica*), but a causative mutation has yet to be found in JQs with PD. Here, we aimed to identify a pathogenic mutation in JQs associated with PD. **Methods**: Paraffin-embedded skeletal muscle blocks from four JQs stored since the 1970s were used in this study. After confirming the histopathological phenotypes of PD, Sanger sequencing was performed to identify a pathological mutation in the *GAA I* gene of JQs. A genotyping survey was conducted using a real-time polymerase chain reaction assay targeting a candidate mutation using DNA samples extracted from 70 new-hatched JQs and 10 eggs from commercial farms. **Results**: Microscopic analysis confirmed the presence of the PD phenotype in three affected JQs based on abnormal histopathological changes and accumulated glycogen in the affected muscles, while one JQ was unaffected and served as a control. Sanger sequencing revealed that the three affected JQs were homozygous for the deletion of guanine at position 1096 in the open reading frame (c.1096delG). A genotyping survey of 70 JQs and 10 eggs from commercial farms showed that none carried this deletion mutation. **Conclusions**: This study identified c.1096delG as the pathogenic mutation for PD in JQs. This mutation induces a frameshift and substitution of amino acids at position 366 (alanine to histidine), resulting in premature termination at the 23rd codon (p.A366Hfs*23). This suggests that this mutation causes the deficient activity of GAA in JQs with PD. The identification of the c.1096delG mutation enabled the systematic maintenance of the flock colony in the PD model. Furthermore, this PD model can be used to clarify unknown aspects of PD pathogenesis and develop therapeutic strategies.

## 1. Introduction

Glycogen storage disease type II (MIM #232300), also known as Pompe disease (PD), is a rare autosomal recessive inherited metabolic disorder caused by a deficiency of lysosomal acid α-1,4-glucosidase (GAA, EC 3.2.1.20), encoded by the *GAA* gene [1]. Impaired GAA activity due to pathogenic mutations in the *GAA* gene results in the abnormal accumulation of glycogen, a substrate of GAA, within intracellular lysosomes, particularly in skeletal and cardiac muscles and other tissues, such as liver cells [2,3]. To date, over 910 variants of the *GAA* gene have been associated with human PD, as documented in the PD *GAA* variant database [4,5].

Human PD is clinically classified into the infantile-onset (IOPD) and late-onset (LOPD) forms [1,6]. IOPD is a rarer and more severe form that typically appears within the first few months of life and is characterized by hypotonia, hypertrophic cardiomyopathy, breathing difficulties, and glycogen accumulation in the cardiac, skeletal, and hepatic tissues, leading to premature death [1,7]. In contrast, LOPD exhibits a heterogeneous presentation. It occurs from childhood to late adulthood and presents with progressive muscle weakness, respiratory insufficiency, elevated levels of creatine kinase, and ptosis [1,7,8]. The involvement of the central nervous system in PD has long been known and is heterogeneous in clinical presentation and magnetic resonance imaging features, occurring in some patients with both IOPD and LOPD [9]. However, several aspects of PD pathogenesis remain unknown, including other genetic factors that determine disease severity and neurological involvement.

Based on the information of the Online Mendelian Inheritance in Animals (OMIA) [10] and the literature [11,12], PD is a multispecies disorder. Cases have been reported in dogs (OMIA 000419-9615) [13], cats (OMIA 000419-9685) [14,15,16], taurine cattle (OMIA 000419-9913) [17], indicine cattle (OMIA 000419-9915) [10,17], sheep (OMIA 000419-9940) [18], and Japanese quail (JQ; *Coturnix japonica*) (OMIA 000419-93934) [19,20,21]. In dogs, a nonsense mutation, c.2237G>A (p.W746*), has been reported in Finnish and Swedish Lapphunds [13]. Three pathogenic mutations: c.1057_1058del (p.Y353L), c.1783C>T (p.R595*), and c.2454_2455del (p.T819R), have been reported in Brahman and Droughtmaster, Brahman, and Shorthorn cattle, respectively [17]. One pathogenic mutation, c.1799G>A (p.R600H), has been reported in a domestic cat [16]. However, to date, no mutations have been identified in sheep or JQ with PD.

A study by a group at the Nippon Institute for Biological Science (NIBS) identified the first suspected PD case in JQ in Japan in 1974 [21,22,23]. A 6-month-old male JQ exhibiting difficulty in raising its wings was observed in a PNN strain flock maintained at NIBS. A closed colony was established for the first case and its offspring, resulting in the establishment of the RWN strain of JQs, which have a defect causing PD in an autosomal recessive manner [21,22,23]. To further examine this JQ PD model, from the 1970s, the NIBS group provided several affected JQs to the Tokyo Metropolitan Institute of Medical Science (TMIMS) [20] and the National Institute of Neuroscience, National Center of Neurology and Psychiatry (NIN-NCNP) [19,24,25,26].

Histopathologically, JQs with PD have a cytoplasm with decreased staining by hematoxylin and eosin (H&E) and cytoplasmic vacuoles in the skeletal and cardiac muscle fibers and liver cells [19,20,21,22]. Periodic acid–Schiff (PAS)-positive cytoplasmic materials, mainly diastase-digestible glycogen, accumulate in the skeletal and cardiac muscles, liver, and, to a lesser extent, the brain, intestine, and gizzard. Biochemically, GAA activity measured using an artificial substrate is markedly reduced, and glycogen accumulates in various tissues of JQs with PD [19,22,25]. Furthermore, an immunoblotting study revealed that the mature form of the GAA protein is deficient in affected JQs [25].

In 1998, the NIN-NCNP group reported two types of JQ *GAA* genes: *GAA I* and *GAA II* [26]. They also demonstrated that the *GAA I* gene is responsible for producing lysosomal GAA activity in JQs and that a lack of *GAA I* mRNA, normally composed of an open reading frame (ORF; 2799 base pairs, 932 amino acids), is a cause of deficient GAA activity and subsequent PD in JQs. However, a causative mutation in the genomic DNA of JQs with PD has yet to be identified.

Therefore, this study aimed to identify pathogenic mutations in JQs with PD using DNA extracted from paraffin-embedded skeletal muscle specimens stored since the 1970s.

## 2. Materials and Methods

### 2.1. Animals and Specimens

We used paraffin-embedded blocks of skeletal muscles from four JQs that had been stored by a researcher (Takane Matsui) at the Obihiro University of Agriculture and Veterinary Medicine since their examination at TMIMS in the 1970s [20,21]. Three specimens were from JQs with PD and one specimen was from the unaffected control JQ. For genotyping, we used 70 frozen new-hatched JQs, which were produced in commercial JQ farms and sold as food for carnivorous wild animals and birds, and the eggshell membranes of 10 JQ eggs, which were sold as human food in supermarkets.

### 2.2. Light Microscopy

Thin sections (4 µm) were prepared from the four paraffin-embedded blocks using a standard method. The sections were stained with H&E and PAS. The histopathological phenotype of PD, including the cytoplasm with decreased H&E staining, cytoplasmic vacuoles in skeletal muscle fibers, and deposition of accumulated glycogen detected by PAS staining [19,20,21,22], was confirmed via light microscopy.

### 2.3. DNA Extraction

For Sanger sequencing and validation of the real-time polymerase chain reaction (PCR) genotyping assay, genomic DNA was extracted from the paraffin-embedded blocks of the three PD-affected and one control JQs. Briefly, several thin sections were prepared from a block, placed in a tube, immersed in 1 mL of xylene, and stirred vigorously. After centrifugation at 10,000× *g* for 2 min at room temperature, the supernatant was removed from the tubes. Next, 1 mL of 100% ethanol was added to the tube, mixed vigorously with the precipitate, and centrifuged at 10,000× *g* for 2 min at room temperature. After the supernatant was removed, the tube was left to stand for approximately 10 min at room temperature with the tube cap open for the vaporization of ethanol. Physiological saline (180 µL) and 20 µL of proteinase K solution (Qiagen Proteinase K, Qiagen, Hilden, Germany) were added to the tube, stirred vigorously, and incubated for 1 h at 56 °C followed by 1 h at 90 °C. After the incubation, genomic DNA was extracted from 200 µL of the final solution using automated extraction equipment (magLEAD 6gC; Precision System Science, Co., Ltd., Matsudo, Japan) and a DNA extraction kit (MagDEA Dx SV; Precision System Science), according to the manufacturer’s recommendations.

For genotyping, genomic DNA was extracted from the pectoral muscles of 70 frozen new-hatched JQs and the eggshell membranes of 10 JQ eggs. An aliquot of new-hatched JQ muscle (approximately 0.1 g) was homogenized with an appropriate amount of sterilized deionized water using an ultrasonic homogenizer (Model UR-20P; Tomy Seiko Co., Ltd., Tokyo, Japan). The homogenate was spotted onto a filter paper (QIAcard FTA Indicating Classic; Qiagen) and stored in a refrigerator (4 °C) until DNA extraction. DNA was extracted from discs punched out of homogenate-spotted filter papers following appropriate treatments, as previously described [27]. An aliquot of an eggshell membrane (approximately 20 mg) was obtained from a JQ egg and cut into small pieces using an 18-gauge needle. DNA was extracted from several pieces of eggshell membrane using a commercial kit (Isohair; Nippon Gene, Tokyo, Japan), according to the manufacturer’s instructions.

### 2.4. Sanger Sequencing

We predicted the structure of the *GAA I* gene of JQs based on the JQ whole genome shotgun sequence (NCBI reference sequence: NC_029529.1; *Coturnix japonica* 2.1) and the JQ *GAA I* mRNA sequence (GenBank accession number: AB000967.1) submitted by the NIN-NCNP group in 1998 [26]. The predicted structure of the *GAA I* gene of JQs is shown in Figure 1 and Appendix A. Based on the predicted structure and sequence of the JQ *GAA I* gene, the exons and exon–intron junctions were amplified and analyzed using Sanger sequencing with specifically designed primer pairs (Appendix A).

PCR was conducted in a 20 µL reaction mixture containing 10 µL 2× PCR master mix (GoTaq Hot Start Green Master Mix, Promega Corp., Madison, WI, USA). The PCR products were purified before sequencing using a QIAquick Gel Extraction Kit (Qiagen), according to the manufacturer’s instructions. Sanger sequencing was performed by Kazusa Genome Technologies Ltd. (Kisarazu, Japan). The obtained sequencing data were analyzed and compared with the reference sequence (NC_029529.1) to identify potential pathogenic mutations.

### 2.5. Real-Time PCR Assay and Genotyping

The primers and TaqMan minor groove binder probes used for the real-time PCR assay (sequences are listed in Appendix A) were designed based on the sequence (NC_029529.1) and a candidate mutation found in this study. These primers and probes, each of which was linked to a fluorescent reporter dye (6-carboxyrhodamine or 6-carboxyfluorescein) at the 5′-end and a non-fluorescent quencher dye at the 3′-end, were synthesized by a commercial company (Applied Biosystems, Foster City, CA, USA). Real-time PCR amplifications were carried out in a final volume of 5 µL consisting of 2× PCR master mix (TaqMan GTXpress Master Mix; Applied Biosystems), 80× genotyping assay mix (TaqMan SNP Genotyping Assays; Applied Biosystems) containing the specific primers at 450 nM, TaqMan probes at 100 nM, and template DNA. A negative control containing nuclease-free water instead of the template DNA was included in each run. The cycling conditions were 20 s at 95 °C, followed by 50 cycles of 3 s at 95 °C and 20 s at 60 °C, with a subsequent holding stage at 25 °C for 30 s. The data obtained were analyzed using StepOne version 2.3 (Applied Biosystems).

DNA samples with two different genotypes, a wild-type homozygote (one JQ) and a mutant homozygote (three JQs), were used to validate the genotyping assay following genotype confirmation based on Sanger sequencing. Genotyping was performed using the newly developed real-time PCR assay with 80 DNA samples obtained from 70 new-hatched JQs and 10 JQ eggs.

## 3. Results

### 3.1. Light Microscopic Features

Light microscopic observations of the H&E-stained specimens from the three JQs with PD revealed that the skeletal muscle fibers had vacuoles and fine granules, and some muscle fibers appeared to be replaced by adipose tissues (Figure 2A,B). In contrast, these characteristics were not observed in the unaffected control JQ (Figure 2C,D). PAS staining detected many glycogen granules in the muscle specimens from the three JQs with PD (Figure 3A,B; Table 1) but not in those from the unaffected control JQ (Figure 3C,D; Table 1). Based on these histopathological findings, we were able to confirm the PD phenotypes of four paraffin-embedded specimens: three affected and one unaffected JQs.

### 3.2. Identification of Variants and a Mutation

Sanger sequencing was performed on 19 exons and exon–intron junctions of *GAA I* in JQs with PD using specific primer pairs (Appendix A). The sequenced exons and exon–intron junctions were compared with the reference sequence (NC_029529.1). Sequencing and comparison revealed six homozygous variants, including five synonymous single-nucleotide substitutions and a guanine deletion at position 1096 (c.1096delG) in the ORF of *GAA I* (Table 2 and Figure 1). These five synonymous variants were identical to those in the normal JQ *GAA I* mRNA sequence (AB000967.1) submitted by the NIN-NCNP group in 1998.

Three JQs with PD were homozygous for the c.1096delG variant (Figure 4). Furthermore, this variant was not present in the sequence of an unaffected control JQ or the normal JQ *GAA I* mRNA sequence (AB000967.1).

### 3.3. Genotyping Survey

The newly developed real-time PCR assay (Appendix A) clearly differentiated the two genotypes, the wild-type homozygote and mutant homozygote, associated with the c.1096delG variant, as evaluated by Sanger sequencing (Figure 4). A genotyping survey using this real-time PCR assay revealed that no JQs in a population of 70 new-hatched JQs and 10 eggs produced in commercial JQ farms carried the c.1096delG variant (Table 3).

## 4. Discussion

Histopathological examinations using H&E and PAS staining confirmed that among the four paraffin-embedded blocks used in this study, three were from JQs affected with PD and one was from an unaffected JQ, with abnormal histopathological changes and accumulated glycogen observed in the muscles of the three affected JQs but not the control (Figure 2 and Figure 3). Sanger sequencing of the DNA samples from the three PD-affected JQs revealed that the pathogenic mutation was c.1096delG (Figure 1 and Figure 4 and Table 2) in exon 6 of the *GAA I* gene, for which all three affected JQs were homozygous, because this mutation was not present either in the sequence of the unaffected control JQ or the reference sequences of the JQ whole genome shotgun sequence (NC_029529.1) and the normal JQ *GAA I* mRNA sequence (AB000967.1). The other five identified variants of a nucleotide substitution were not pathogenic, because they were all synonymous alterations that induced no amino acid substitutions (Table 2) and were also present in the normal JQ *GAA I* mRNA sequence (AB000967.1).

The c.1096delG mutation induces a frameshift and substitution of amino acids at position 366 (alanine to histidine) and onward, eventually, a premature termination at the 23rd codon (p.A366Hfs*23) (Figure 4). In all eukaryotes, there is a quality-control mechanism called nonsense-mediated mRNA decay (NMD), which selectively degrades mRNAs carrying premature terminations or nonsense codons to prevent them from producing truncated proteins with deleterious gain-of-function or dominant-negative activity [28,29]. Therefore, the JQ *GAA I* mRNA carrying the c.1096delG (p.A366Hfs*23) mutation was likely eliminated by the NMD mechanism. Indeed, the NIN-NCNP group previously showed that no *GAA I* mRNA was detected in any tissue from JQs with PD in both reverse transcription-PCR and northern blotting experiments [26]. These observations strongly suggest that this deletion mutation, which induces a premature termination codon, is responsible for the deficient lysosomal GAA activity in JQs with PD.

In the process of establishing RWN strain at NIBS, a flock colony of JQs with PD, backcross mating was performed using the founder (first case, male, 6 months old in 1974) and the first and second generations of JQs [21,22,23]. Wing movement abnormalities were first found in a 3-month-old female JQ in 1977 and then in two male JQs (7 and 8 months old) in 1978. Thereafter, the NIBS group continued to select the JQs with wing movement abnormalities and low GAA activity. Symptoms associated with PD appeared mostly in aged males at the beginning of selection; however, it tended to develop in younger JQs and in both sexes after the third generation. Consequently, the NIBS group established the RWNE strain, which develops wing movement abnormalities before 6 weeks of age [23]. From this early onset RWNE strain, the NIBS group isolated JQs with late-onset PD and established the RWNL strain, which developed abnormalities after 4 months of age [22,23].

In human PD, a comparison of IOPD and LOPD genotypes revealed that homozygosity was predominantly found in IOPD cases and compound heterozygosity in LOPD cases [30]. According to this observation in human PD, JQs in the RWNE strain might exhibit PD symptoms through homozygosity of the c.1096delG mutation identified in this study, whereas heterozygosity of the deletion mutation and another milder mutation might cause late-onset PD in the RWNL strain. Alternatively, a modifier associated with early- or late-onset PD may have caused differences in the age at onset and severity of PD in the JQs. However, further research is required to confirm these hypotheses.

Enzyme replacement therapy (ERT) has remained the standard treatment for human PD since 2006 and has significantly extended the survival of both children and adults with PD [31]. ERT successfully prolongs the survival of patients with IOPD and stabilizes disease progression in patients with LOPD by ameliorating cardiac dysfunction [12,31,32]. However, its effect on the respiratory phenotype is limited. While ERT for PD continues to improve following enhancements in recombinant human GAA, integrated approaches that combine different therapeutic strategies, including gene therapy, substrate reduction therapy, chaperone therapy, inhibition of autophagy, stimulation of lysosomal exocytosis, antisense oligonucleotides, and genome editing are expected [12,31]. Although the usefulness of the JQ PD model has declined in recent years, its historical contribution is significant due to it being the first animal model to be used for testing ERT [33]. In this study, we successfully identified a pathogenic mutation in JQs with PD; therefore, this avian PD model may be valuable for emerging therapeutic strategies, such as gene therapy, antisense oligonucleotides, and genome editing.

## 5. Conclusions

This study identified c.1096delG as the pathogenic mutation responsible for PD in JQs. This mutation induces a frameshift and substitution of amino acids at position 366 (alanine to histidine), leading to a premature termination at the 23rd codon (p.A366Hfs*23), suggesting that this deletion mutation is responsible for the deficient activity of GAA in JQs with PD. The c.1096delG mutation and the newly developed real-time PCR assay will enable the systematic maintenance of flock colonies of this avian PD model. Furthermore, the JQ PD model can be utilized to clarify unknown aspects of PD pathogenesis and develop therapeutic strategies, although further research is required to confirm the genetic characteristics of the existing flocks of the JQ PD model before its utilization.

## Figures and Tables

**Figure 1 genes-16-00975-f001:**
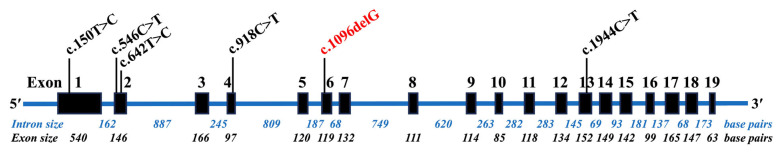
Predicted structure of the *GAA I* gene of Japanese quails. The locations of all exons and introns were determined based on the whole genome shotgun sequence of Japanese quails (NCBI reference sequence: NC_029529.1; *Coturnix japonica* 2.1) and the *GAA I* mRNA sequence of Japanese quails (GenBank accession number: AB000967.1).

**Figure 2 genes-16-00975-f002:**
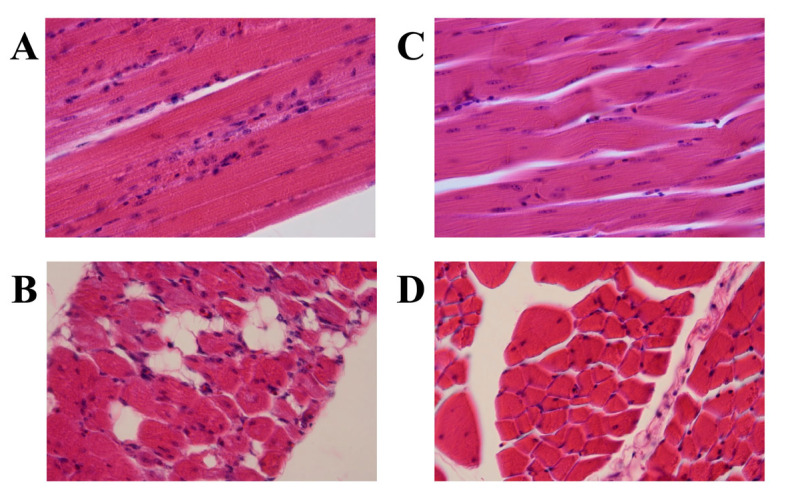
Histopathological changes in the skeletal muscle from a Japanese quail (JQ) with Pompe disease (PD) compared with that from an unaffected control JQ. (**A**) Longitudinal and (**B**) cross-sectional views in affected JQ and (**C**) longitudinal and (**D**) cross-sectional views in unaffected JQ. Hematoxylin and eosin staining. Muscle fibers have vacuoles and fine granules (**A**), and some muscle fibers appear to be replaced by adipose tissues (**B**) in the JQ with PD but not in the control JQ (**C**,**D**).

**Figure 3 genes-16-00975-f003:**
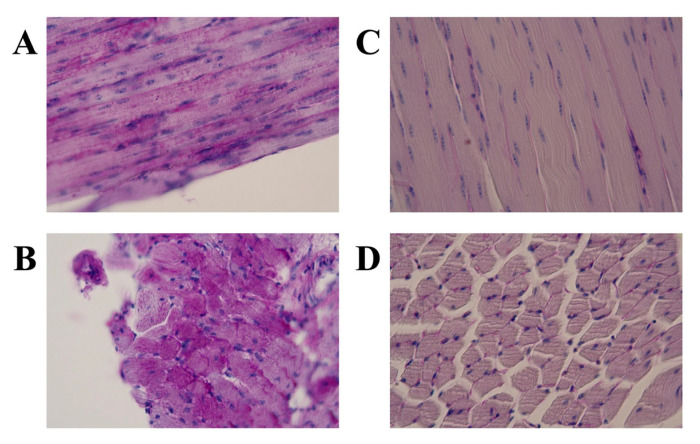
Glycogen accumulation detected with periodic acid–Schiff staining in the skeletal muscle from a Japanese quail (JQ) with Pompe disease (PD) compared with that from an unaffected control JQ. (**A**) Longitudinal and (**B**) cross-sectional views in affected JQ and (**C**) longitudinal and (**D**) cross-sectional views in unaffected JQ. Periodic acid–Schiff staining. Muscle fibers have many positively-stained glycogen granules in the JQ with PD (**A**,**B**) but not in the control JQ (**C**,**D**).

**Figure 4 genes-16-00975-f004:**
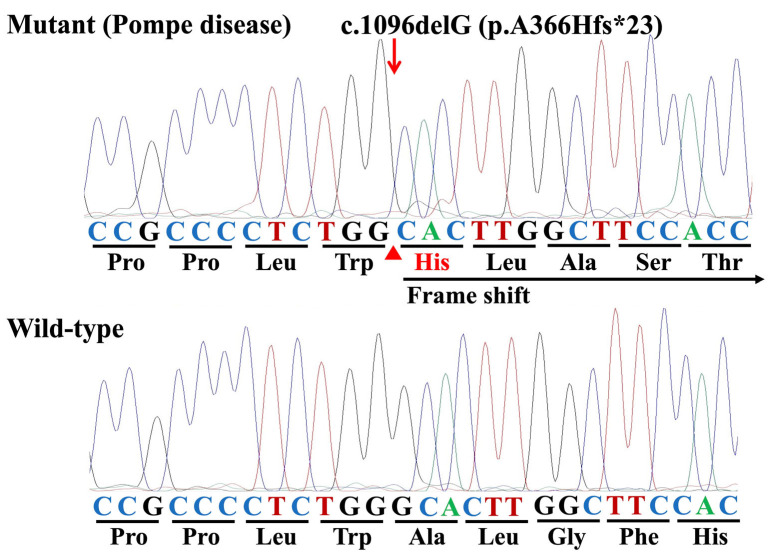
Representative Sanger sequencing electropherograms illustrating the mutant homozygote (Pompe disease) and the wild-type homozygote associated with the c.1096delG (p.A366Hfs*23) in the *GAA I* gene of Japanese quails (arrow).

**Table 1 genes-16-00975-t001:** Glycogen accumulation detected with periodic acid–Schiff (PAS) staining in the skeletal muscle from Japanese quails (JQs) with Pompe disease (PD) and an unaffected control JQ.

JQ	Glycogen Accumulation	
	Positive *	Negative **
PD	3 (100%)	0 (0%)
Control	0 (0%)	1 (100%)

* PAS staining detected many glycogen granules in the muscle specimens, and ** PAS staining did not detect glycogen granules in the muscle specimen.

**Table 2 genes-16-00975-t002:** Identified variants in the *GAA I* gene of Japanese quails with Pompe disease.

Exon	Variant (Nucleotide) *	Variant (Amino Acid)	Interpretation
Exon 1	c.150T>C	p.T50T	synonymous
Exon 2	c.546C>T	p.T182T	synonymous
Exon 2	c.642T>C	p.D214D	synonymous
Exon 4	c.918C>T	p.H306H	synonymous
Exon 6	c.1096delG	p.A366Hfs*23	deletion (deleterious)
Exon 13	c.1944C>T	p.S648S	synonymous

* Variants were determined by comparison with a reference sequence (NCBI reference sequence: NC_029529.1; *Coturnix japonica* 2.1).

**Table 3 genes-16-00975-t003:** Genotyping survey in a population of 70 new-hatched Japanese quails (JQs) and 10 eggs from commercial JQ farms.

JQ	Number Examined	*GAA I*:c.1096delG		
		Mutant Homozygote	Heterozygote	Wild-Type Homozygote
New-hatched JQ	70	0 (0%)	0 (0%)	70 (100%)
Egg	10	0 (0%)	0 (0%)	10 (100%)

## Data Availability

The original contributions presented in this study are included in the article/Appendix A. Further inquiries can be directed to the corresponding author.

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
