# Peer review of "Identification of a Pathogenic Mutation for Glycogen Storage Disease Type II (Pompe Disease) in Japanese Quails (*Coturnix japonica*)"

_genes, 2025, doi:10.3390/genes16080975_

Round 1

Reviewer 1 Report

Comments and Suggestions for Authors

The aim of the study was to identify a pathogenic mutation in Japanese quails associated with Pompe disease. The materials used for the study were taken from the quails and the eggs shell and compare with the tissues stored in the paraffin-embedded skeletal muscle blocks from four quails stored since the 1970s. Firstly, confirmation of the histopathological phenotypes of Pompe disease was done and secondly, Sanger sequencing was performed to identify a pathological mutation in the GAA I gene of quails. After that genotyping survey was conducted using a real-time polymerase chain reaction assay targeting a candidate mutation using DNA samples extracted from 70 newborn quails  and 10 eggs from commercial farms. The results confirmed the existence of the mutation in the three fragments of muscles stored in the paraffin blocks and excluded in the tissues taken from chicks and eggs shell. The quails model is useful for the checking the existence of mutated genes.

The methods were nvel, adequate to the aim of the study, presentation of the results and discussion as well as conclusion were logical and scientific.

The only suggestion is to better description of the method of DNA isolation from paraffin blocks.

Author Response

Responses to the comments from Reviewer 1

Dear Reviewer 1,

First of all, we appreciate the time the editor and reviewers have taken to read and review our manuscript. Their valuable comments have significantly improved several aspects of our paper. The following document presents our responses to comments and suggestions from the reviewer. It includes the original comments in italics and blue and the subsequent responses we made. The revised parts are indicated in red.

Comments and Suggestions for Authors

The aim of the study was to identify a pathogenic mutation in Japanese quails associated with Pompe disease. The materials used for the study were taken from the quails and the eggs shell and compare with the tissues stored in the paraffin-embedded skeletal muscle blocks from four quails stored since the 1970s. Firstly, confirmation of the histopathological phenotypes of Pompe disease was done and secondly, Sanger sequencing was performed to identify a pathological mutation in the GAA I gene of quails. After that genotyping survey was conducted using a real-time polymerase chain reaction assay targeting a candidate mutation using DNA samples extracted from 70 newborn quails and 10 eggs from commercial farms. The results confirmed the existence of the mutation in the three fragments of muscles stored in the paraffin blocks and excluded in the tissues taken from chicks and eggs shell. The quails model is useful for the checking the existence of mutated genes.

Authors’ response: We thank the reviewer for understanding the objectives and strengths of our study. We appreciate it.

The methods were novel, adequate to the aim of the study, presentation of the results and discussion as well as conclusion were logical and scientific.

Authors’ response: We thank the reviewer for highly evaluating our data in this study. We appreciate it.

The only suggestion is to better description of the method of DNA isolation from paraffin blocks.

Authors’ response: We thank the reviewer for this suggestion. That is probably that this description is a relatively long and too detailed. We described a kind of detail procedure about an extraction of DNA from paraffin-embedded blocks because this procedure has not been described before. We believe that this procedure is a key in this study, which is able to efficiently extract DNA that was damaged and fragmented due to a long storage from the 1970s. Everyone can reproduce this experiment according to this described procedure. We appreciate the reviewer’s understanding. Furthermore, we added a DNA extraction kit used in this procedure, which we forgot to write as follows.

 We will carefully revise our paper according to the reviewer's comments and suggestions as follows.

“After the incubation, genomic DNA was extracted from 200 µL of the final solution using automated extraction equipment (magLEAD 6gC; Precision System Science, Co., Ltd., Matsudo, Japan) and a DNA extraction kit (MagDEA Dx SV; Precision System Science) according to the manufacturer’s recommendations.”

Reviewer 2 Report

Comments and Suggestions for Authors

Identification of a Pathogenic Mutation for Glycogen Storage Disease Type II (Pompe Disease) in Japanese Quails (Coturnix japonica)

Dear Authors,

The manuscript is interesting and well-prepared. Locating the mutation will certainly allow for monitoring flocks using real-time PCR assays. Further studies would certainly require a larger number of birds. Maybe it may also be possible to sample a larger area of the country, based on stratification criteria, and, taking into account the percentage of farms and bird numbers in a given area, proportionally sample a larger area. what important entire mechanism can improve level of knowledge about PD in case of human newborns, in case of humans frequency of disease is 1:18,711 (https://doi.org/10.3389/fped.2023.1221140). Main problem in my opinion is the lack of a precisely described form of data analysis, even in the form of simple tables and frequencies with one additional subsection. Besides of this, information about paraffin-embedded skeletal muscle blocks is mentioned, that is why the experiment did not require ethical committee approval, but only possible approval from the appropriate unit on Department responsible for animal welfare.

Below I added some suggestions helpful in revision process:

Lines 19-20

In this part of manuscript is emphasized that PD was reported in several animals and Japanese quails. Maybe better in future is to increase number of animals/muscle blocks in analysis to 1000 animals/50 blocks or even more (it depends from analysis costs) because in case lower number of animals this kind of analysis can have higher power of a test (additionally when statistical comparison will be added in form even contingency tables with numbers of observations and frequencies  in form of percentages), and let more precise determined percent of animals in population, but that is also involved with sampling (stratification of samples from entire population of JQ’s).

Lines 26, 129, 131, 174

New-hatched will be more appropriate description than new-born (placenta) in case of JQ’s.

Lines 102-106

It is worth emphasizing that studies determining the presence of glycogen in muscle tissue were performed on animal material rather than on live animals and did not require the approval of the Ethics Committee. Furthermore, it is also worth emphasizing the approval of the relevant Departmental Committee for Animal Welfare/similar unit at Obihiro University of Agriculture and Veterinary Medicine based on the relevant regulation (No....).

Line 106

Samples from 70 new-hatched JQ’s and 10 eggs can be also emphasized in subsection 2.1.

Line 181

One additional short subsection 2.6. Data analysis/ 2.6. Data validation can be added to emphasized simple descriptive statistics and data gathered in form of tables describing frequency (number of observations and percentages).

Lines 183-191

Simple table emphasized frequency of data in terms of number of positive and negative cases of glycogen accumulation in classes can be added.

I.e.:

Table 1.

Item

Glycogen accumulation

positive

negative

Control

-

1 (100%)

Blocks with PD

3 (100%)

-

Line 225-230

Simple table emphasized frequency of data in terms of number of positive and negative cases of identified mutation in GAA I gene in classes can be added.

Table 3.

Item

GAA I mutation

positive

negative

New-hatched JQ’s

-

70 (100%)

Eggs

-

10 (100%)

Author Response

Responses to the comments from Reviewer 2

Dear Reviewer 2,

First of all, we appreciate the time the editor and reviewers have taken to read and review our manuscript. Their valuable comments have significantly improved several aspects of our paper. The following document presents our responses to comments and suggestions from the reviewer. It includes the original comments in italics and blue and the subsequent responses we made. The revised parts are indicated in red.

Comments and Suggestions for Authors

Identification of a Pathogenic Mutation for Glycogen Storage Disease Type II (Pompe Disease) in Japanese Quails (Coturnix japonica)

Dear Authors,

The manuscript is interesting and well-prepared. Locating the mutation will certainly allow for monitoring flocks using real-time PCR assays. Further studies would certainly require a larger number of birds. Maybe it may also be possible to sample a larger area of the country, based on stratification criteria, and, taking into account the percentage of farms and bird numbers in a given area, proportionally sample a larger area. what important entire mechanism can improve level of knowledge about PD in case of human newborns, in case of humans frequency of disease is 1:18,711 (https://doi.org/10.3389/fped.2023.1221140). Main problem in my opinion is the lack of a precisely described form of data analysis, even in the form of simple tables and frequencies with one additional subsection. Besides of this, information about paraffin-embedded skeletal muscle blocks is mentioned, that is why the experiment did not require ethical committee approval, but only possible approval from the appropriate unit on Department responsible for animal welfare.

Authors’ response: We thank the reviewer for highly evaluating our study, pointing out our future tasks of molecular epidemiology, and suggesting some weak points including data presentation and ethical committee approval. We revised our paper carefully according to the reviewer's comments and suggestions regarding the sample number and ethical committee approval as follows and also explained the reasons in detail in case we were not able to revise in certain issues including data presentation.

Below I added some suggestions helpful in revision process:

Lines 19-20

In this part of manuscript is emphasized that PD was reported in several animals and Japanese quails. Maybe better in future is to increase number of animals/muscle blocks in analysis to 1000 animals/50 blocks or even more (it depends from analysis costs) because in case lower number of animals this kind of analysis can have higher power of a test (additionally when statistical comparison will be added in form even contingency tables with numbers of observations and frequencies in form of percentages), and let more precise determined percent of animals in population, but that is also involved with sampling (stratification of samples from entire population of JQ’s).

Authors’ response: We thank the reviewer for this valuable suggestion. We totally agree with the reviewer’s opinion. We think that the result in this study is an initiation of our quail PD research. As we already mentioned in the Conclusion section, we will analyze the existing flock of quail PD and survey enough number of quails from farms in Japan and possibly other countries. We appreciate it.

Lines 26, 129, 131, 174

New-hatched will be more appropriate description than new-born (placenta) in case of JQ’s.

Authors’ response: We thank the reviewer for the valuable advice. We came to know that it is right and changed the word “newborn” to “new-hatched” at all five places in our paper. We appreciate it.

Lines 102-106

It is worth emphasizing that studies determining the presence of glycogen in muscle tissue were performed on animal material rather than on live animals and did not require the approval of the Ethics Committee. Furthermore, it is also worth emphasizing the approval of the relevant Departmental Committee for Animal Welfare/similar unit at Obihiro University of Agriculture and Veterinary Medicine based on the relevant regulation (No....).

Authors’ response: We thank the reviewer for pointing this out. We added a phrase about this issue in the Institutional Review Board Statement section as follows.

Institutional Review Board Statement: This study was conducted in accordance with the Guidelines for Regulating Animal Use and Ethics of Kagoshima University (approval no. VM15041; approval date: 29 September 2015) and the Animal Experiment Committee of Obihiro University of Agriculture and Veterinary Medicine (approval no. 25-106; approval date: 17 March 2025).

Line 106

Samples from 70 new-hatched JQ’s and 10 eggs can be also emphasized in subsection 2.1.

Authors’ response: We thank the reviewer for point this out. We described these samples in subsection 2.1. that was moved from subsection 2.2., and deleted the repeated description from the second paragraph of subsection 2.2. We appreciate it.

Line 181

One additional short subsection 2.6. Data analysis/ 2.6. Data validation can be added to emphasized simple descriptive statistics and data gathered in form of tables describing frequency (number of observations and percentages).

Lines 183-191

Simple table emphasized frequency of data in terms of number of positive and negative cases of glycogen accumulation in classes can be added.

Line 225-230

Simple table emphasized frequency of data in terms of number of positive and negative cases of identified mutation in GAA I gene in classes can be added.

Authors’ response: We really thank the reviewer for the valuable comments and recommendations. We well understand the reviewer’s goodwill to improve our paper. We really appreciate it. However, we did not revise our paper in respect of this issue. So, we would like to explain the reasons as follows.

In this study, we used all the stored paraffin-embedded blocks: three PD affected and one unaffected control quails from the previous PD-experimental flock in the 1970s. We did not choose these samples randomly from the stored blocks of quails with unknown diseases. We first reconfirmed the histological phenotypes in the four PD samples and analyzed molecularly using Sanger sequencing. Eventually, we successfully found the causative mutation that was deleterious enough to be pathogenic because of a guanine deletion leading to a frameshift and correlated with a PD phenotype (an accumulation of glycogen). We do not think that statistics is needed for the demonstration of this result because these samples were not chosen randomly.

In addition, we preliminarily surveyed molecularly using 70 new-hatched quails and 10 eggs to confirm that the deletion mutation was not easily found in normal quails produced from farms. We do not think that statistics and presentation using a table are necessary in these data. As the reviewer above-mentioned, this scale of survey is not enough to determine the precise mutant allele frequency of the normal population in farms. Furthermore, it is necessary to check molecularly the existing PD-related flock to find another possible molecular basis that may determine an infantile or late onset. For the sake of these future surveys and examinations, we need the first demonstration of the mutation causing PD, i.e., publication in a trustworthy journal like Genes. Immediately after this publication, we can contact the farms and institutes for further researches of quail PD. We can analyze statistically the data obtained in the near future. We appreciate the reviewer’s understanding.